# Free Radical Inhibition Using a Water-Soluble Curcumin Complex, NDS27: Mechanism Study Using EPR, Chemiluminescence, and Docking

**DOI:** 10.3390/antiox13010080

**Published:** 2024-01-08

**Authors:** Ange Mouithys-Mickalad, Koffi Senam Etsè, Thierry Franck, Justine Ceusters, Ariane Niesten, Hélène Graide, Ginette Deby-Dupont, Charlotte Sandersen, Didier Serteyn

**Affiliations:** 1Centre for Oxygen R&D (CORD)-CIRM, Institute of Chemistry, University of Liège, Allée de la Chimie, 3, 4000 Liège, Belgium; t.franck@uliege.be (T.F.); j.ceusters@uliege.be (J.C.); ariane.niesten@uliege.be (A.N.); fa511444@skynet.be (G.D.-D.); didier.serteyn@uliege.be (D.S.); 2Laboratory of Medicinal Analytic (CIRM), University of Liège, Hospital Quarter, 15 Hospital Avenue, 4000 Liège, Belgium; ks.etse@yahoo.com; 3Veterinary Clinic, Large Animal Surgery, B32, Boulevard du Rectorat, 4000 Liège, Belgium; charlotte.sandersen@uliege.be

**Keywords:** curcumin, NDS27, HPβCD, HRP, superoxide anion, singlet oxygen, EPR, chemiluminescence, docking

## Abstract

There is a growing interest in the use of natural compounds to tackle inflammatory diseases and cancers. However, most of them face the bioavailability and solubility challenges to reaching cellular compartments and exert their potential biological effects. Polyphenols belong to that class of molecules, and numerous efforts have been made to improve and overcome these problems. Curcumin is widely studied for its antioxidant and anti-inflammatory properties as well as its use as an anticancer agent. However, its poor solubility and bioavailability are often a source of concern with disappointing or unexpected results in cellular models or in vivo, which limits the clinical use of curcumin as such. Beside nanoparticles and liposomes, cyclodextrins are one of the best candidates to improve the solubility of these molecules. We have used lysine and cyclodextrin to form a water-soluble curcumin complex, named NDS27, in which potential anti-inflammatory effects were demonstrated in cellular and in vivo models. Herein, we investigated for the first time its direct free radicals scavenging activity on DPPH/ABTS assays as well as on hydroxyl, superoxide anion, and peroxyl radical species. The ability of NDS27 to quench singlet oxygen, produced by rose bengal photosensitization, was studied, as was the inhibiting effect on the enzyme-catalyzed oxidation of the co-substrate, luminol analog (L012), using horseradish peroxidase (HRP)/hydrogen peroxide (H_2_O_2_) system. Finally, docking was performed to study the behavior of NDS27 in the active site of the peroxidase enzyme.

## 1. Introduction

Curcumin (1,7-bis-(4-hydroxy-3-methoxyphenyl)-1,6-heptadiene-3,5-dione) is a natural phenolic compound used as the main ingredient found in curcuma longa root [1]. Curcumin is also well-known as an antioxidant and anti-inflammatory agent, acting as a protector against intracellular reactive oxygen species (ROS) [2]. The chemical and biochemical mechanisms by which curcumin exerts antioxidant and protective actions have been widely investigated mainly against the oxidation of biomolecules [3,4] and lipid oxidation [5]. Importantly, the precise role of phenol and enol groups on the antioxidant mechanism is now well known [6]. On the other hand, much attention has been devoted to the in vivo effect of curcumin, with discrepancies in the results depending on the redox potential of the cellular compartment targeted and the bioavailability of the molecule [7]. Indeed, curcumin has very poor solubility in aqueous mediums, which results in a low availability that hinders the study of its properties in in vivo models. Targeting oxidative stress in vivo means knowing the main sources of free radicals/reactive oxygen species and their tight mechanisms of reaction.

The NADPH oxidase (NOX-2) found in the polymorphonuclear neutrophils (PMN) as well as in monocytes/macrophages [8] is the main provider of superoxide anion (O_2_^•−^), the primary free radical from which the other species derive [9,10]. Still, when looking at cellular depth, the mitochondrion is considered the producer and target of free radicals, making it an interesting therapeutic target to control its dysfunction and modulate the production of free radicals and ROS within the mitochondrial compartment [11]. Moreover, a growing interest has been focused on mitochondria-targeted polyphenols with the idea of intervening in the levels of ROS/free radicals in mitochondria [12]. Polyphenols are more than only antioxidants, and studies of their interactions with the organelles or the products of their metabolism need to be well understood. Polyphenols can act either as anti- or pro-oxidants, as well as inhibitors or enhancers of oxidant stress and radical processes [13]. The redox potential of the polyphenol and the system under investigation, thus, appear to be crucial for biological properties.

As a result, growing interest has been focused on finding new formulations of curcumin that have similar chemical and pharmacological properties. Consecutively, several groups have synthesized nanoparticles [14,15], liposomes [16,17], and complexes including cyclodextrins containing the parent molecule, curcumin [18,19,20]. Amongst those complexes formed by encapsulation within a cyclodextrin macromolecule, a new water-soluble form of curcumin has been developed in our group named NDS27. It is obtained by adding curcumin-lysine to hydroxypropyl-beta-cyclodextrin (HP-beta-CD or HPED, Figure 1) [21].

We have already demonstrated that NDS27 can keep its antioxidant and anti-inflammatory properties on redox processes involving two enzymes of neutrophil: NADPH-oxidase and myeloperoxidase (MPO), which have been studied in cell-free systems [22], specifically, on cellular ROS production [23], as well as in a model of induced neutrophilia in horses [24]. 

Altogether, our previous results indicate that the new curcumin formulation (NDS27) could act on radical species via an electron exchange; however, we have no proof that it turns itself into a radical state, which has not yet been directly demonstrated even for its parent molecule, curcumin [25]. The structure requirement for antioxidant activity seems to be the presence of a hydroxyl group [6] attached to the aromatic ring, which can be transformed into a phenoxy radical, favoring the free radical scavenging activity [26]. The methoxy and α,β-unsaturated carbonyl moiety could also contribute to the beneficial effects of curcumin. In these conditions, the presence of cyclodextrin in the complex (NDS27) may influence its reactions. Herein, we have investigated the effect of NDS27 on free radicals such as ABTS, DPPH, superoxide anion, hydroxyl, and lipid peroxyl radicals to gain more information on the precise mechanism of action of the NDS27 and on the appearance of the phenoxy radical from curcumin by using direct electron paramagnetic resonance (EPR) spectroscopy combined with a spin trapping technique. Note that, apart from the EPR method used in this study to evidence the free radical scavenging activity, nuclear magnetic resonance (NMR) can be employed as an alternative method to verify the quality of dietary supplements containing curcumin [27], mainly when working with the sample coming from natural extracts. In addition, the ability of NDS27 to quench singlet oxygen (^1^O_2_), produced using rose bengal photosensitization, was also studied and compared to its parent molecule curcumin using optical spectroscopy. The inhibiting effect on the enzyme-catalyzed oxidation of the substrate, a luminol analog, L012, using horseradish peroxidase (HRP)/hydrogen peroxide (H_2_O_2_) system was completed with docking to study the behavior of NDS27 into the active site of the HRP enzyme. 

## 2. Materials and Methods

### 2.1. Chemicals and Reagents

All the salts used to prepare the buffered solutions were purchased from Merck (VWR, Leuven, Belgium) and were of analytical grade. The chemiluminescent probe, L012 (8-amino-5-chloro-7-phenyl-pyrido[3,4-d]pyridazine-1,4(2H,3H)dione), was obtained from Wako Chemical Europe (Neuss, Germany); high-purity 5,5′-dimethyl-pyrroline-N-oxide (DMPO) was from Alexis Biochemicals via Enzo Life Sciences (Brussels, Belgium). 4-POBN (4-pyridyl-1-oxide)-*N-tert*-butylnitrone), DPPH (2,2′-diphenyl-1-picrylhydrazyl), ABTS (2,2′-azino-bis(3-ethylbenzothiazoline)-6-sulfonic acid), potassium superoxide (KO_2_), xanthine oxidase from bovine milk, linoleic acid, Chaps (3-[(3-Cholamidopropyl)dimethylammonio]-1-propanesulfonate hydrate), vitamin C, sodium azide (NaN_3_), and iron (II) sulfate heptahydrate (Fe_2_SO_4_, 7H_2_O), were all from Sigma–Aldrich (Steinheim, Germany). Horseradish peroxidase (HRP), superoxide dismutase (SOD), ethylene diamine tetraacetic acid disodium (EDTA), hypoxanthine, and diethylene triamine pentaacetic acid (DTPA) were from Merck (Darmstadt, Germany). Curcumin (98.2% HPLC) was from PalChem (Angers, France). Hydroxypropyl-β-cyclodextrin (HPβCD or HPED) was obtained from Roquette (Lestrem, France). Curcumin-lysine, NDS27 (complex containing curcumin-lysine + hydroxypropyl-beta-cyclodextrin), was obtained from Bioptis (Vielsam, Belgium). The linoleic acid emulsion was prepared as previously described [28]. Methanol, ethanol, dimethyl sulfoxide (DMSO), and hydrogen peroxide (H_2_O_2_) were of analytical grade from Merck VWR (Leuven, Belgium). Quercetin and resveratrol were both from Fluka (Buchs, Switzerland). Anthracene-9,10-dipropionic acid disodium salt (ADPA) and rose bengal (RB, 2,3,4,5-tetrachloro-6-(3-hydroxy-2,4,5,7-tetraiodo-6-oxo-xanthen-9-yl)benzoic acid) were from Sigma–Aldrich (Steinheim, Belgium). Percoll was from GE Healthcare (VWR, Leuven Belgium). Bovine serum (BSA) and phorbol 12-myristate 13-acetate (PMA) were from Sigma–Aldrich (Bornem, Belgium). Cell culture medium (IMDM) was from Lonza (Verviers, Belgium), Hank’s balanced salt solution (HBSS), and additives (fetal bovine serum (FBS), amphotericin, antibiotics) were from Sigma–Aldrich (Steinheim, Belgium). Sterile culture flasks were from Greiner Bio-One (Vilvoorde, Belgium), and white microplate and transparent plates were from Fisher Scientific (Tournai, Belgium).

### 2.2. Preparation of the Solutions

The stock solutions of the tested compounds (NDS27, curcumin-lysine, curcumin, and sodium azide) were prepared as follows: for the curcumin complex (NDS27), 79.59 mg of powder was weighed and dissolved in 1 mL of ultra-pure distilled water to obtain a stock solution of 10^−2^ M, with dilutions of 5·10^−3^, 10^−3^, 5·10^−4^, and 10^−4^ M. Similarly, the stock solution (5·10^−2^ M) of the excipient hydroxypropyl-β-cyclodextrin (HPED or HPβCD) was prepared in distilled water (dH_2_O) and further diluted to obtain 5·10^−3^, 10^−3^, 5·10^−4^, 10^−4^, and 5·10^−3^, 10^−3^, and 5·10^−4^ M. Curcumin and its curcumin-lysine salt were both prepared at 10^−2^ M by dissolving in DMSO. Appropriate dilutions were then performed to obtain 5·10^−3^, 10^−3^, 5·10^−4^, 10^−4^, and 10^−5^ M, etc. To obtain the final concentrations in the reaction vials, each tested compound was diluted 100 times (or 10 times when required) with the appropriate solvent (DMSO 1% or H_2_O) for all the samples at the final dilution. For the other reference molecules such as quercetin or resveratrol, the stock solution was also made at 10^−2^ M in DMSO with dilutions of 5·10^−3^, 10^−3^, 5·10^−4^, and 10^−4^ M.

### 2.3. Cell Culture

Human promyelocytic leukemia cells (HL-60) were obtained from the American Type Culture Collection (ACCT, USA) and cultured in Iscove’s Modified Dulbecco’s Medium (IMDM) supplemented with 20% (*v*/*v*) fetal bovine serum, 100 U/mL penicillin/streptomycin, 1.25 µg/mL amphotericin B, and 2 g/L NaHCO_3_ in 50 mL flasks at 37 °C in a 5% CO_2_ humidified atmosphere. The cells were cultured and fed two to three times per week to maintain a log phase growth; moreover, once a week, they were centrifuged and re-suspended in fresh IMDM. Before each experiment, cells were counted with Burker cells (Briare, France) to reach the cell density required for the experiments, i.e., 1·10^6^ cells/mL.

#### 2.3.1. Measurement of the ROS Produced by PMA-Activated HL-60 Monocytes

This experiment was performed based on the method previously described by Amado et al. [29] with slight adaptations [28,30]. One million HL-60 monocytes were plated in a 48-well microplate and incubated at 37 °C with 5% CO_2_ in a humidified atmosphere. The content of every three wells was transferred into a 5 mL tube for centrifugation (300× *g*) for 10 min at 37 °C. The recovered cell pellet was diluted with 500 μL HBSS buffer and transferred into wells containing 450 μL HBSS. Ten microliters of compound (NDS27 or curcumin or curcumin-lysine), corresponding to the final concentrations ranging from 5·10^−8^ M to 5·10^−6^ M, were added to the cell suspension (950 µL) and completed using 5 μL HRP (30 μg/mL) and 30 μL PMA (0.486 μM) to trigger the cell activation. For the excipient HPED, the concentrations’ range was 10^−6^ M to 10^−4^ M. In each assay, three wells were loaded without a sample and were taken as a positive control for ROS production (Ctrl A). To measure the basic ROS production of HL60 cells in the absence of activation, three other wells without a tested compound received no PMA (Ctrl NA, not activated). The third control was subjected to the presence of distilled water or DMSO (Ctrl dH_2_O, or Ctrl DMSO). The chemiluminescence (CL) was measured in the presence of 10^−4^ M L012 for 30 min at 37 °C on a fluorescent microplate reader (Fluoroskan Ascent FL, Fischer Scientific, Tournai, Belgium).

#### 2.3.2. Isolation of Equine Neutrophils (PMN) and ROS Production Using Stimulation with PMA

We implemented another experiment using neutrophils which undergo respiratory bursts when stimulated, producing superoxide anion radicals. The radical interaction with NDS27 can be evidenced by using electron paramagnetic resonance (EPR).

Equine neutrophils were isolated from whole blood using EDTA disodium salt (1.6 mg/mL) as an anticoagulant. The blood was drawn from the jugular vein of healthy horses bred and fed under identical conditions and without medical treatment. All the experiments were carried out with the approval of the ethics committee of the Faculty of Veterinary Medicine of the University of Liege (agreement number 1474). Briefly, the neutrophils were isolated at room temperature (18–22 °C) using centrifugation (400× *g*; 45 min; 20 °C) on a discontinuous Percoll density gradient according to the method previously described [31,32]. The cells were gently collected and washed with two volumes of physiological saline solution (PBS). After the supernatant removal, the cell pellets were re-suspended in 2 mL PBS and counted for further use. Ten microliters of compound (NDS27), corresponding to the final concentrations of 10^−4^ M and 10^−3^ M, were added to the cell suspension. The ROS were produced following PMN activation (6·10^6^/mL) with 30 μL PMA (0.486 μM) in a total volume of 200 µL PBS (pH 7.4).

### 2.4. Free Radical Scavenging Activity (ABTS and DPPH)

#### 2.4.1. ABTS Test (EPR Assay)

The ABTS assay was based on the method described by Re et al. [33]. For the generation of ABTS^•+^ radicals, an aqueous solution of 7 mM ABTS was added to the buffered solution of HRP. The reaction was triggered by adding 1 mM H_2_O_2_ in a vial and mixing vigorously to obtain a green-colored solution as previously described, with slight modification [34]. In this assay, the HRP (100 µg/mL)/H_2_O_2_ (1 mM) in the phosphate buffer (pH 7.4) was used instead of metmyoglobin (MetMb)/H_2_O_2_. The resulting solution of ABTS^•+^ was directly introduced into the capillary and transferred to the EPR tube for analysis. For the assay, the tested compound (NDS27) was assayed at 10^−5^ M and 10^−4^ M, and at 5·10^−4^ M for the excipient HPβCD (or HPED). Curcumin and the salt (curcumin-lysine) were both tested at 10^−4^ M. A control consisted of 2 µL of DMSO or ultra-pure water in 200 µL of buffered solution (1%, *v*:*v*). During this reaction, the blue-green ABTS^•+^ radical cation is converted back into its colorless neutral form in the presence of the potential antioxidant molecule (e.g., curcumin or NDS27). Reducing capacity was determined using a comparison of the EPR signal intensity of the sample versus the positive control. All the EPR spectra were recorded on a Bruker spectrometer (EMX-micro, Karlsruhe, Germany) at room temperature with the following experimental settings: microwave frequency (9.8 GHz); Center field (3480 G) microwave power (18 mW); sweep width (20 G); modulation frequency (100 kHz), and the total number of scans was 4.

#### 2.4.2. DPPH Test (Optical and EPR Assays)

The DPPH assay was performed according to the method developed by Brand-Williams et al. [35] albeit slightly modified [30]. A solution of 1 mM DPPH in methanol (ethanol for EPR) was stirred for 40 min. The absorbance of the solution was adjusted to 0.650 ± 0.020 at 517 nm using fresh methanol (or ethanol for EPR). Absorption spectra were recorded at room temperature with a multi-channel (1024-element diode array) Agilent 8453 spectrophotometer (Hewlett–Packard, Waldbronn, Germany), and the samples were placed in a quartz cell, with an optical path of 1 cm. For EPR, 2 µL of curcumin, used as a reference, or NDS27 was mixed with 198 µL of DPPH solution and automatically analyzed using EPR spectroscopy in the EPR cell tube at room temperature. When reacting with an antioxidant, the DPPH^•^ radical is converted into DPPH, and its color changes from purple to yellow. The antioxidant effect was evaluated by observing the decrease in EPR signal intensity over time with EMX micro-Bruker spectroscopy. A control consisted of 2 or 20 µL of DMSO or ultra-pure water in 198 or 190 µL of DPPH solution. The DPPH radical scavenging activity was determined by measuring the EPR signal intensity of the sample versus the positive control (DPPH radical in EtOH). The experimental settings were like those used for the ABTS test (see above). The concentrations’ range for the EPR assay was 10^−4^ and 10^−4^ M for NDS27 and 5·10^−5^ and 10^−4^ M for curcumin. The excipient was tested at 10^−4^ M. For the optical study, the final concentration for all the samples ranged from 10^−6^ M to 10^−4^ M.

### 2.5. Enzyme-Catalyzed Reactions (Optical, CL, and EPR Studies)

The horseradish peroxidase (HRP) reacts with hydrogen peroxide (H_2_O_2_), yielding a “compound I [HRP^+^-Fe(IV=O)]”, which is the reactive intermediate of HRP able to oxidize any potential reducing agent. The effect of NDS27, or parent molecule (curcumin), on the HRP activity was monitored by using L012-enhanced chemiluminescence and EPR in combination with a spin trapping technique (with DMPO as spin trap agent).

#### 2.5.1. Chemiluminescence Study of Curcumin and NDS27 on Enzyme-Catalyzed Oxidation of L012

The direct effect of NDS27 and that of its parent molecule (curcumin) were studied on the peroxidase (HRP)-catalyzed oxidation of the chemiluminescent probe (L012), in the presence of hydrogen peroxide (H_2_O_2_).

Briefly, 2 μL of tested compound (NDS27), or reference molecules (curcumin, resveratrol, quercetin), were incubated in phosphate buffer at pH 7.4 (153 μL, 50 mM) containing 5 μL HRP (100 µg/mL), 20 μL L012 (10^−3^ M), and 20 μL H_2_O_2_ (1 mM); immediately, the chemiluminescence was measured using a fluorescent plate reader (Fluoroskan Ascent FL, Fischer Scientific, Tournai, Belgium) at 37 °C for 30 min. All the assays were performed in triplicate and repeated at least twice. The relative activity of the tested compound or reference (curcumin or resveratrol/quercetin) was compared to control in the presence of DMSO used as a solvent or distilled water used to prepare the stock solution of NDS27 and the excipient (HPβCD = HPED). All the tested compounds were used at the final concentration ranging from 12.5 µM to 62.5 µM.

#### 2.5.2. EPR Study of Curcumin and NDS27 on the Enzyme-Catalyzed Oxidation Reaction

To investigate the direct effect of NDS27 and its parent molecule (curcumin) on the reactive intermediates of enzyme peroxidase (HRP), EPR spectroscopy was performed in phosphate buffer (pH 7.5) at room temperature with or without a combination of the spin trapping technique (DMPO). Consequently, 10 µL HRP (3 mg/0.5 mL, stock solution), 10 µL NDS27 (10^−2^ M, stock solution), 60 µL buffer, and 10 µL DMPO (100 mM, final concentration) were mixed, and the reaction was triggered by adding 10 µL H_2_O_2_ (1 mM). The microcapillary containing the tested solution was transferred to a quartz cell and placed into the EPR cavity. All the EPR spectra were recorded on an EMX-micro Bruker spectrometer with the following experimental settings: microwave frequency (9.8 GHz); center field (3480 G) microwave power (18 mW); sweep width (100 G or 20 G); modulation frequency (100 kHz), and the total number of scans was 4 or 6.

### 2.6. In Vitro Production of Superoxide Anion Radical and Hydroxyl Radical

Superoxide anion was produced chemically by KO_2_ or enzymatically using hypoxanthine/xanthine oxidase while hydroxyl radical was produced by the Fenton reaction. For the inhibitory action on the superoxide anion radical, NDS27 or the excipient (HPED), or its parent molecule (curcumin), EPR, in combination with a spin trapping technique (with DMPO as spin trap agent) was employed.

#### 2.6.1. Production of Superoxide Anion (O_2_^−^) Using KO_2_

To produce the superoxide anion (O_2_^−^) chemically, potassium superoxide (KO_2_) powder was used and dissolved either in the methanol—ultra-pure water (60:40, *v*:*v*) mixture or in a buffer on ice at a concentration of 1 M and diluted afterward to reach a final concentration of 1 mM. The reactivity of the superoxide anion radical and the curcumin complex (NDS27) was monitored by using an optical absorption technique in the quartz cuvette (2 mL volume).

For the EPR study, the final volume was 100 µL, containing 60 µL of MeOH/H_2_O or phosphate buffer (pH 7.4), 30 µL of KO_2_ (0.1 M), and 10 µL of DMPO (1 M). For the sample, 2 µL of curcumin (5·10^−4^ and 10^−3^ M), NDS27, or HPED as excipient both at 10^−3^ M were added. Superoxide dismutase (SOD) was assayed at 10 and 20 U/mL. Another experiment was performed with the superoxide anion prepared with deuterated water (D_2_O) before adding it to the phosphate buffer (pH 7.4), see Appendix A.

#### 2.6.2. In Vitro Production of Superoxide Anion Using the Hypoxanthine/Xanthine Oxidase System: Detection Using EPR Technique and L012 Enhanced Chemiluminescence

Superoxide anion (O_2_^−^) was produced from the activity of xanthine oxidase (20 U) as previously described [31,36] with slight modifications and by using hypoxanthine instead of xanthine in the phosphate buffer (pH 7.8, 50 mM). The stock solution of hypoxanthine (10 mM) was prepared in an aqueous solution after adding droplet alkaline (NaOH, 1 M). For the EPR assay, the effect of NDS27, at 10^−5^ M and 2·10^−5^ M on the formation of superoxide anion in this model, was compared to its excipient cyclodextrin (HPED, at a final concentration of 5·10^−5^ M). For chemiluminescence assays, all the samples were tested at the final concentrations of 10^−6^, 5·10^−6^, 10^−5^, and 5·10^−5^ M at 37 °C for 30 min.

The EPR assay was carried out in the presence of 100 mM DMPO. The experimental settings were quite identical to that used for the reaction with KO_2_ except that the number of scans was 4.

#### 2.6.3. In Vitro Production of Hydroxyl Radical Using Fenton Reaction: Iron (II)Sulfate/H_2_O_2_: EPR Technique

The Fenton reaction was conducted as previously described [31] albeit slightly modified, namely, in ultra-pure water in the presence of 10 mM DMPO, 2.5·10^−4^ M DTPA, 1 mM H_2_O_2_, and 5·10^−4^ M iron (II), which was used to trigger the reaction. The experimental setup used a center field of 3481.6 G, sweep width of 100 G, attenuation of 10 dB, modulation frequency of 100 kHz, modulation amplitude of 1 G, receiver gain of 2·10^4^, power of 18.85 mW, time constant of 40.96 ms, conversion time of 20.0 ms, and a total number of 2 scans. Similar final concentrations of 10^−5^ M and 2·10^−5^ M were used for NDS27 and 5·10^−5^ M for HPED.

### 2.7. Production of Singlet Oxygen upon Irradiation of Rose Bengal

Singlet oxygen (^1^O_2_) was produced upon the irradiation of the photosensitizer, rose bengal (RB), in the presence of ADPA, an anthracene derivative, whose absorbance at 400 nm decreased over time in phosphate buffer (pH 7.4, 50 mM) at room temperature [37]. The stock solutions of RB (10^−4^ M) and ADPA (10^−3^ M) were prepared in distilled water and phosphate buffer (pH 7.5), respectively. Upon irradiation, RB produces ^1^O_2_ with a known quantum yield (0.76) as it strongly absorbs around 550 nm [38]. The tested compounds (NDS27, HPβCD, and curcumin) were compared to a known ^1^O_2_ quencher, sodium azide, while DMSO and distilled water were used as solvents, respectively. The compound of interest, NDS27 or curcumin, was assayed at a final concentration of 5·10^−6^ M and 5·10^−5^ M and compared to sodium azide at the final concentration ranging from 5·10^−5^ M to 5·10^−6^ M. The excipient was tested at 5·10^−5^ M.

### 2.8. EPR Study on Lipid Peroxidation Model

Preparation of the linoleic acid emulsion: Linoleic acid (0.64 mM) and 200 mg CHAPS were mixed in 50 mL chelexed phosphate buffer (pH 7.4) and gently shaken for 5 min at room temperature. Afterward, the suspension was filtered and stored till further use. The peroxidation was triggered by adding iron (II) in the presence of ascorbate (vitamin C).

#### Measurement of the Transient Free Radicals Produced by Lipid Peroxidation (EPR Technique)

The study was performed according to the method described by [28]. Briefly, 100 µL of tested compound was added to the final concentrations ranging from 5·10^−5^ M to 5·10^−4^ M for NDS27 or 5·10^−5^ M to 10^−3^ M for curcumin and added to 500 μL of linoleic acid emulsion (0.32 mM) before the addition of 100 μL of vitamin C (10^−3^ M) and 100 μL of POBN (50 mM). The excipient HPED was tested at 5·10^−5^ M and 5·10^−3^ M. For controls, samples were carried out with solvent instead of the compound of interest and taken as 100% production of transient free radicals. Each tube volume was completed to 1 mL of total volume with chelexed phosphate buffer (pH 7.4) and 10 μL of (5·10^−5^ M) Fe(II) that were added last. The tube was stored in the incubator at 37 °C in the dark for 2 h and the content was transferred to an EPR quartz flat cell that was placed into the cavity of the spectrometer for analysis. The experimental parameters were as described in Materials and Methods (Section 2.4.2), except that the spin trap agent was POBN instead of DMPO.

### 2.9. Molecular Docking Study

Molecular docking calculations were performed with AUTODOCK 4.2 [39] according to the procedure we recently reported, with slight modifications [30]. The crystal structure of HRP (PDB code 1HCH) with a resolution of 1.57 Å retrieved from the protein data bank (PDB) was used in the docking study [40]. All the water molecules were removed, missing hydrogen atoms were added, and non-polar hydrogens were merged into their corresponding carbons using AUTODOCK. A grid box size of 35 × 35 × 35 Å^3^, with a spacing of 0.375, was centered on the HRP heme. The 3D molecular structure of the curcumin used for the docking was optimized using the GAMESS interface at the B3LYP 3-21G level of the theory. During the docking simulation, 10 conformers were first considered by using the default genetic algorithm for each run. Finally, 100 runs were performed to obtain a robust population for the analyses. Input preparation was performed by using MGLTools-1.5.6. Further, AUTODOCK was used to create a PDBQT file and generate the Grid Parameter File (GPF). The grid box parameters were also saved in GPF file format, and the Grid Log File (GLG) was generated by running an AUTOGRID protocol. Then, the Docking Parameter File (DPF) was generated with the LAMARCKIAN genetic algorithm by setting the protein as a rigid molecule. Finally, the DLG (Docking Log File) file is created by running AUTODOCK with DPF as an input file. Using the Autogrid module, the electrostatic and atomic interaction maps for all atoms of curcumin were calculated. Out of the several interactions possible, the best were those with high interaction, the lowest binding energy, and ligand efficiency; these were considered for further HRP–Curcumin docking discussion. PyMOL, version 2.4.0, was used for docking conformation representation [41].

### 2.10. Statistical Analysis

Data are given in absolute values in reference to control distilled water (dH_2_O) for NDS27 and its excipient HPED or DMSO for curcumin and the salt (curcumin-lysine). All data are expressed as the mean +/− SD of at least two independent experiments performed in triplicate. A one-way ANOVA with a DUNNET test was performed. *p* < 0.05 was considered significant and NS was non-significant.

## 3. Results

### 3.1. Measurement of Cellular ROS Produced by Monocytes and Neutrophils

#### 3.1.1. Effect of NDS27 on ROS Produced by Monocytes Compared to Curcumin and Curcumin-Lysine: Chemiluminescence Study

The effect of this new water-soluble curcumin complex (NDS27) on the ROS produced by the cellular model (monocytes line) was compared to its precursor (curcumin salt) and the parent molecule (curcumin) to determine if the encapsulated form maintains antiradical activity. Figure 1 shows that a dose-dependent inhibitory action on the light emission was observed but was less pronounced than that for its precursor salt (curcumin-lysin or LysCurc) and curcumin. The excipient HPED at various concentrations did not impact the ROS production compared to the positive control (activated HL60 cells: Ctrl A). Note that DMSO (Ctrl A+ DMSO), used to solubilize curcumin, and the salt (LysCurc) already inhibited the ROS production, increasing the effect of the curcumin and the salt.

#### 3.1.2. Inhibiting Effect of NDS27 versus Curcumin on the O_2_^•−^ Generated by PMA-Stimulated Neutrophils: EPR Study

The activity of NDS27 on superoxide anion radicals produced by stimulated neutrophils, evidenced by EPR-spin trapping in the presence of DMPO as a spin trap, has shown a strong inhibiting effect at 10^−4^ M and total disappearance of the EPR spectrum at the high concentration of 10^−3^ M (Figure 2C,D); meanwhile, the excipient hydroxy-β-cyclodextrin, HPED at 10^−3^ M, had only a weak non-significant effect (Figure 2B). 

### 3.2. Free Radicals Scavenging Activity (ABTS/DPPH)

#### 3.2.1. ABTS Results

Using direct EPR spectroscopy, curcumin, and curcumin-lysine were inhibited by 33 and 36% of the formation of ABTS^•+^ radicals (Figure 3, spectra C, D) in comparison with DMSO (Figure 3, spectrum B). However, this inhibition lowered to about 17 and 21%, respectively, compared to the positive control (Figure 3, spectrum A). When NDS27 was used instead of curcumin, a similar inhibition (16%) was observed versus positive control (Figure 3, spectra F and G compared to spectrum E). At the highest concentration of 10^−4^ M, this effect was pronounced, reaching 46% inhibition.

#### 3.2.2. DPPH Results

##### Optical Study of the DPPH Test

Through an optical study, it appears that NDS27, prepared in distilled water, exhibited a moderate and quite identical scavenging activity to the salt (curcumin-lysine) at the two highest concentrations while its parent molecule curcumin, prepared in DMSO, showed a pronounced action (Appendix A). When quercetin used as a standard was tested, a strong scavenging activity towards DPPH radicals was observed that was more pronounced than curcumin. The excipient cyclodextrin (HEPD = HPβCD), used as the vehicle of NDS27, remained inactive (Appendix A).

##### DPPH Direct EPR Study

To investigate in depth the free radical-inhibiting mechanism of NDS27, a direct EPR study was performed without the addition of the spin trap agent. The EPR spectrum was a five-line EPR signal, characteristic of the DPPH radical [42], which was not modified in the presence of HPED even at the high concentration of 10^−4^ M (Figure 4A). Upon the addition of increasing concentrations of NDS27 (10^−4^ and 2·10^−4^ M), a decrease in EPR signal intensity was seen and was even transformed into a single-line EPR spectrum when the concentration of NDS27 reached 2·10^−4^ M, which was attributed to the curcumyl radical. Likewise, the same results were obtained with quite similar concentrations of curcumin (Figure 4B).

### 3.3. Enzyme-Catalyzed Reactions

#### 3.3.1. Effect of NDS27 on the Enzymatic System (HRP/H_2_O_2_) Optical Spectroscopy Studies

The conversion of horseradish peroxidase (HRP), in the presence of the natural substrate H_2_O_2_, showed the formation of intermediate species named Cpd II, Cpd III, and Cpd P-670 in the phosphate buffer (pH 7.4) through optical spectroscopy (Appendix A). In our experimental conditions, the highly reactive intermediate (cpd I) was not seen. Appendix A shows the influence of NDS27 on the peroxidase activity over time. It appears that NDS27 slightly increased the formation of Cpd II and even Cpd III (for Appendix A).

#### 3.3.2. Effect of NDS27 on the Enzyme System (HRP/H_2_O_2_): Using Chemiluminescence (CL) Assay

In comparison to the chemical assays (DPPH), our results obtained with the enzyme model (HRP/H_2_O_2_) using CL technique, in the presence of L012 as a probe, show that most of the tested molecules exhibited a variable dose-dependent effect on light emission (Appendix A). NDS27 exhibited a nice inhibiting effect compared to the excipient (HPβCD = HPED) which was found ineffective in the system. In this model, curcumin showed a strong inhibitory action that was more pronounced than NDS27 on the emitted light (Appendix A).

Based on the CL results, it seemed interesting to investigate this reaction using EPR spectroscopy, the unique technique dealing with electron and paramagnetic species.

#### 3.3.3. Effect of NDS27 on Enzymatic System (HRP/H_2_O_2_): EPR Study

The peroxidase activity was monitored using an EPR technique, with or without the use of a spin trapping agent (DMPO) (Figure 5). In the absence of DMPO, no EPR signal was detected; meanwhile, a quadruplet of doublet EPR spectrum, corresponding to a π–cation radical intermediate trapped by DMPO, was seen in spectrum B. When 10^−3^ M NDS27 was added to the complete system (in the presence of DMPO), a single EPR signal appeared, centered at 3480 G, likely due to the formation of the curcumyl radical (Figure 5, spectrum C). To obtain more information, an additional experiment was performed with a higher concentration (10^−3^ M) of NDS27 without DMPO (Figure 5, spectrum D): a similar signal was observed, attributed to curcumyl radical, suggesting a radical transfer to the curcumin.

### 3.4. In Vitro Production of Superoxide Anion and Hydroxyl Radical

#### 3.4.1. Chemical Production of Superoxide Anion: Optical Results for KO_2_ + NDS27 and Beta-Cyclodextrin (HPED)

The absorbance peak of the curcumin complex (NDS27) showed a main peak located at 436 nm with a shoulder at a lower wavelength of 340 nm (Figure 6, red line). The addition of 1 mM KO_2_ in the mixture induced a bathochromic shift (489 nm) (Figure 6, yellow line).

#### 3.4.2. EPR Study

Figure 7 shows the reaction between NDS27, and the superoxide anion radical produced in the MeOH-H_2_O (60/40, *v*:*v*) mixture or in the phosphate buffer (pH 7.4) at room temperature was monitored using EPR spectroscopy (Figure 7). The EPR spectrum of DMPO-^•^OOH adducts, which is the pattern of the superoxide anion trapped by DMPO, showed a 10-line EPR signal (Figure 7A). The excipient HPED at 10^−3^ M was without effect (Figure 7E), and SOD at increasing concentrations (10, 20, and 30 U), as expected, had an inhibiting action on the EPR signal (spectra B–D).

NDS27 at 10^−3^ M inhibited the EPR signal (Figure 7I) more than the curcumin did at 10^−3^ M (Figure 7F). Curcumin and curcumin-lysine (Figure 7, spectra F, and H) both showed a similar effect on the superoxide anion but were less pronounced than NDS27.

Appendix A shows the EPR spectra obtained when O_2_^•−^ was produced in deuterated water (D_2_O), and the experiment was performed in phosphate buffer (pH 7.4) (spectrum A). NDS27 at increasing concentrations (5·10^−5^ M, 5·10^−4^ M, and 10^−3^ M) showed a dose-dependent decrease, with total disappearance of the six-line spectrum at 10^−3^ M (Appendix A, spectra F–H).

#### 3.4.3. Production of Superoxide Anion Using Enzymatic Pathway (Hypoxanthine/Xanthine Oxidase)

NDS27 inhibited the EPR signal produced by the hypoxanthine/xanthine oxidase system in a dose-dependent manner (Figure 8d,e). The control with SOD at 20U completely abolished the EPR signal intensity (spectrum c). The excipient HPED exhibited a very weak effect (Figure 8, spectrum b).

By using a chemiluminescence technique, a similar dose-dependent inhibitory action of the O_2_^•−^ radical was observed for NDS27 with comparative efficacy for its parent molecule curcumin and the curcumin-lysine (Appendix A).

#### 3.4.4. Effect of NDS27 on Hydroxyl Radical Produced the Fenton Reaction

The hydroxy radical generated by the Fenton reaction characterized by a four-line EPR spectrum (trapped by DMPO, a_N_ = a_H_ = 14.9 G) was dose-dependently reduced using NDS27 (26 and 47%, respectively, for 10^−5^ M and 2·10^−5^ M) (Figure 9, spectra c and d). Cyclodextrin (HPED) at 5·10^−5^ M showed a weak inhibitory action (24%) versus the positive control (spectrum b).

### 3.5. Singlet Oxygen-Quenching Effect of NDS27

Upon the irradiation of the photosensitizer, rose bengal (RB), in the presence of the anthracene derivative (ADPA), a slow and continuous decrease in the absorbance is observed over time; meanwhile, in the absence of RB, this decrease in absorbance was strongly slowed down (Appendix A). Moreover, in this model, distilled water and DMSO were used as solvents, and HPED was used as an excipient but had a very weak or no quenching effect on ^1^O_2_ generation (Appendix A). When the reaction was performed in the presence of a well-known singlet oxygen inhibitor, azide, a nice dose-dependent slowdown in the formation of ^1^O_2_ was observed (Appendix A). Interestingly, when curcumin or NDS27 was used instead of azide, a similar slowdown in the formation of singlet oxygen was seen as shown in Appendix A).

### 3.6. Curcumin and NDS27 Behavior during Lipid Peroxidation

#### EPR Spin Trap Study: Lipid Radical Scavenging Effect of NDS27 versus Curcumin

The results are summarized in Figure 10. In the absence of the ascorbate/Fe(II) system, the lipid peroxidation did not start (Figure 10A, spectrum b). In contrast, the attack of the linoleic acid emulsion by the triggering system (ascorbate/Fe(II)) in the presence of the spin trap agent (POBN) led to an EPR spectrum of high intensity (Figure 10A, spectrum a), which characterizes the trapping of free radicals by POBN [43]. Compared to the control (Figure 10A, spectrum a), the different solvent controls (with distilled H_2_O, DMSO, and HPED) (Figure 10A, spectra c–f) did not have an impact on the EPR signal’s intensity, even at the highest concentration of HPED (spectrum f). On the contrary, the addition of NDS27 or curcumin, at increasing final concentrations of 5·10^−5^ M, 10^−4^ M, and 5·10^−4^ M, resulted in a decrease in the intensity of the EPR signal in a concentration-dependent manner (Figure 10B, spectra a–f) when compared to the complete-system positive Ctrl (spectrum Aa). NDS27 showed a significant inhibitory effect vs. the controls (DMSO, dH_2_O, and HPED).

### 3.7. Molecular Docking Study of Curcumin (NDS27)-HRP Complex

The docking results show that curcumin occupies two privileged positions in the binding cavity, called “*E*” and “*I*”, corresponding to the molecular pose at the entrance of the cavity and inside the cavity, respectively. Figure 11a,b shows the two privileged positions of the curcumin inside the HRP binding pocket. Altogether, these poses form a “nipper-like” shape around the protoporphyrin heme. Whatever the considered group, one phenyl ring is positioned up, and the second phenyl down the heme (Figure 11c). The ketone carbonyl groups located between the two phenyl groups are oriented toward the outside of the heme. The best poses of each privileged position and the interactions observed are then analyzed.

Compounds of category “*E*” interact with the binding domain thanks to hydrogen bonds (HBs) as well as through hydrophobic interaction. The hydrogen atom of the phenol group positioned under the heme interacts with one oxygen atom of the carbonyl group of Asp247 and Ser167. Furthermore, the hydrogen atom at the *alpha* position of this phenol oxygen on the aromatic ring is oriented toward one oxygen of the Asp247 (Figure 11d). For the linker between the phenol rings of curcumin, results showed that one of the ketone oxygen atoms interacts thanks to HB with the residue Ser151. Concerning the aromatic ring placed above the heme, an HB between the phenol oxygen atom and the N-*H* hydrogen of His42 is suggested. The same oxygen atom is positioned near the N-*H* hydrogen of Arg38, leading to an additional HB.

For the docking solution of the “*I*” category, differences in the binding mode with the HRP active site are observed. Although interaction with Arg38 is also observed, such contact is realized with another hydrogen atom of the residue and the oxygen of the curcumin phenol. The O-*H* hydrogen of the phenol group placed up the heme interacts both with the Ser73 oxygen and the oxygen atom of the propanoic acid side chain of the heme. This last interaction is interesting since it could induce efficient structural change close to the heme. The effect of that contact is probably due to the best ligand efficiency value of the “*I*” category pose (20.7) contrary to that of category “*E*” (27.0). The other end phenol group of the curcumin, positioned under the heme, shows proximity and, therefore, interaction with His170 (Figure 11e). Whatever the pose, in addition to the interactions highlighted above and the hydrophobic contacts, the phenyl ring placed up the heme could interact with the imidazole rings of the protoporphyrin thanks to π···π stacking. A combination of all these poses with close contact with the HRP binding domain residues could justify the activity observed in the experimental assay.

## 4. Discussion

There is a growing interest in the use of natural molecules to avoid the side effects observed when classical medicines are employed. On the other hand, much attention has been devoted to the in vivo effect of polyphenol derivatives like curcumin, with discrepancies in the results. The latter is mainly dependent on the redox potential found within the cellular compartment targeted and on the bioavailability of the molecule [7]. Numerous groups have developed new pharmaceutical preparations to improve the capacity of the molecule of interest [19,44]. Curcumin has been used for a long time to fight several inflammatory diseases including cancer, especially when associated with another anti-cancer treatment [45]. There are few adverse effects known due to curcumin per se. Nevertheless, curcumin and other compounds belonging to the polyphenol family suffer from poor solubility and low bioavailability, which hinder their use in a clinical context. If an increase in bioavailability is suitable to reach different cellular compartments, it can also be as harmful as other toxic components brought by nutriments or other ingredients that might easily enter.

Different forms were prepared for this purpose, such as liposomes, cyclodextrins, and nanoparticles [14,46,47]. More importantly, nanoparticles of curcumin were comparatively more effective than the native curcumin against different cancer cell lines. Amongst those approaches, the use of cyclodextrins to encapsulate drugs is very interesting, as this type of excipient, approved by the EU and USA regulatory agencies, is compatible and could improve the potential therapeutic effect of the drug [48]. That is why cyclodextrins are suited for oral administration. We have chosen this approach to bio-optimize our curcumin preparation containing lysine.

This new complex named NDS27 has been tested on in vitro-like in vivo models to verify its efficacy [23,24]. Considering the similarity of equine and human inflammatory lung disease, our group successfully conducted a study with NDS27, achieved through inhalation at 100 mg twice daily to treat equine asthma by using an equine model of neutrophilia induced by LPS inhalation versus the excipient HPED alone [24]. Likewise, we also investigated the effect of this formulation on the equine neutrophil activation and degradation with quite a better efficacy on the ROS and myeloperoxidase activity [23]. However, the precise mechanism of action on ROS of this new curcumin formulation is not fully elucidated. In cellular and in vivo assays, it is not excluded, that during the reaction with ROS or on the ROS-producing enzyme, the excipient could also participate, mainly by acting on lipid membranes and other components of the cells. This cyclodextrin behavior has been already described in the literature [48]. Most of the reactions of phenolic compounds are based on the radical exchange, following the electron transfer (ET) and/or hydrogen atom transfer (HAT) process. The chemical structure of curcumin offers different sites of attack. The possible site for free radical oxidants on the curcumin is the diketone, which can be transformed into the phenoxy intermediate (on the phenol group). The delocalization of the electron can stabilize the curcumyl radical and be detected by an accurate technique such as EPR. However, curcumin can also be regenerated in the presence of reducing agents like ascorbate acid which will exchange the electron from the transient species of phenoxy radicals. The study of ROS, produced in the cellular model, implies the activated species and the radical ones. The techniques often used in these conditions only target the global reactive oxygen species, without distinguishing their radical nature. Therefore, EPR spectroscopy, a sensitive and accurate technique, is a highly suitable tool of choice to evidence the presence of free radicals.

First, using a chemiluminescence technique, we have confirmed the capacity of NDS27 and curcumin to inhibit the ROS produced by monocytes (HL60 cells) through an NADPH oxidase pathway, which agrees with our previous findings [31] and those from others [49]. The NDS27 effect on light emission is mainly due to curcumin present in the complex because the excipient HPED did not modify the CL. The strong inhibition seen for curcumin and its salt curcumin-lysine can be attributed in part to the residual effect of the solvent used (DMSO) [50]. In this case, the tested molecule, or DMSO, competes with O_2_^•−^ radical and the CL enhancer, L012. Because of the very weak EPR signal obtained with activated HL-60 cells, the assay was performed with neutrophils.

Neutrophils (PMNs), through NOX activity, are known to produce a high amount of O_2_^•−^ radicals during the respiratory burst [51]. It was important to study how the new curcumin form, NDS27, reacts with O_2_^•−^ by using the specific technique (EPR), in combination with spin trapping, to easily detect the free radicals. The curcumin salt encapsulated in β-cyclodextrin and its effect can also arise from the excipient. It has been reported that cyclodextrins, due to their properties, might be used for specific medicinal purposes [48]. Through this methodology, we have demonstrated that NDS27 can scavenge the superoxide anion radical and lower the resulting EPR signal intensity. The EPR patterns (aN = 14.9 G and aHb = 12.7 G) are characteristic of the DMPO-^•^OOH adduct of the superoxide anion [52]. As cells are complex, an additional experimental setup was designed to study the direct interaction of NDS27 and the parent molecule, curcumin. It consists of the production of superoxide anion radical, either through the chemical decomposition of potassium dioxide (KO_2_) or enzymatically using hypoxanthine/xanthine oxidase. In both in vitro acellular models, we have shown that NDS27 interacts with O_2_^•−^ by lowering the EPR signal, as does superoxide dismutase. As the O_2_^•−^ can undergo rapid dismutation, leading to hydrogen peroxide, which triggers the formation of another highly reactive species, hydroxyl radical, the effect of the tested molecules, appears moderate but is interesting for NDS27, mainly at the highest concentration of 10^−3^ M (Figure 7). To obtain more information, the same experiment was performed in the deuterated buffer to lower this O_2_^•−^ self-dismutation [53].

It turns out that the ability of NDS27 to scavenge the superoxide anion is enhanced: at 10^−3^ M, it completely suppresses the EPR signal (Appendix A).

Another way to produce O_2_^•−^ is by using the enzymatic system (hypoxanthine/xanthine oxidase). The first step was to follow the ROS generation using chemiluminescence enhanced by the L012 probe (see Figure 2). A dose-dependent inhibition of the chemiluminescence was observed for NDS27 but was more pronounced for curcumin and curcumin-lysine, while the excipient remained inactive. Using the EPR technique, a similar dose-dependent action was also observed on the DMPO-^•^OOH adduct compared to the positive control. Interestingly, the excipient HPED did not show any effect on the superoxide anion in both techniques. The NDS27 effect on the HO^•^ radical can be partially explained by a chelating effect of the parent molecule, curcumin, as already described [54].

Taken together, our results demonstrate that the main mechanism of action for curcumin and its water-soluble form (NDS27) remains throughout the radical transfer, as evidenced by the efficacy of the latter one on the Fenton reaction, producing the hydroxyl radical, which mainly reacts through the radical process. In the living system, there is another highly reactive species that participates in oxygen toxicity. Singlet oxygen is one of the harmful species for healthy cells but can be also used to fight cancer cells. On the other hand, it seems that the use of curcumin in the treatment of cancer, even if widely reported, remains for many clinicians an achievable therapeutic approach because there are no real clinical trials proven to be successful in humans [55]. However, its additional effect when used in association with anti-cancer drugs is widely documented [56,57]. It is known that curcumin can react with singlet O_2_; however, to our best knowledge, there are no studies dealing with the reactivity of a new encapsulated curcumin complex with ^1^O_2_.

Therefore, it is interesting to know if the curcumin in the β-cyclodextrin keeps the capacity to react with this species. We have used a spectroscopic model to produce and evidence the singlet oxygen based on our previous works [37]. The rose bengal photosensitization leads to ^1^O_2_ production, which reacts with the anthracene derivative (ADPA), yielding the endoperoxide derivative. We have demonstrated that NDS27 reacted with ^1^O_2_ like curcumin. Again, as with previous assays, the excipient does not impact this reaction, even at the highest concentration of 10^−3^ M, indicating that only the active ingredient, curcumin, causes this quenching effect. Likewise, the use of azide, a well-known ^1^O_2_ quencher, shows a dose-dependent slowdown of the kinetics of singlet oxygen production. We did not observe in our conditions the formation of a novel peak corresponding to the formation of oxidized curcumin, likely due to the overlap at the wavelength studied. Based on the literature, curcumin can react with ^1^O_2_ and yield other derivatives [58]. The discrepancy between the two results can be explained by the presence of a cyclodextrin moiety that slowly releases outside the active ingredient.

The production of ROS and free radicals is particularly important in the development of some diseases where the cellular membranes are affected notably by their lipid components. Hydroxyl radicals and singlet oxygen are both implicated in lipid peroxidation [49,59]. There is no study dealing with the effect of the new water-soluble curcumin against lipid radicals derived from lipoperoxidation systems. By using our previously described model [28], we have shown that curcumin and NDS27 reduced, in a dose-dependent manner, the lipid peroxidation, as evidenced by the data obtained from the EPR spin trapping technique. This result agrees with those obtained in different models where superoxide anion, hydroxyl radical, and singlet oxygen are produced. This also indicates that the ET and/or HAT mechanisms predominate as a mode of action of NDS27. However, for singlet oxygen, the quenching effect cannot be excluded due to the dual action of curcumin [60].

We have seen that NDS27 can react mainly through a radical process, and it was also important to complete the mechanistic pathway by studying the radical scavenging properties. We used the chemical classical test (DPPH) and produced the ABTS^•+^ radical by using an enzymatic system (HRP/H_2_O_2_). In both models, an inhibitory action was observed for NDS27 like with curcumin. These findings indicate that NDS27 acts through two radical processes, as evidenced by direct EPR spectroscopy, with the formation of curcumyl radical located, even temporally, on the structure of curcumin (the phenoxy radical or the carbon-centered one). Interestingly, this radical species was also observed for the first time when the encapsulated derivative reacted with the stable DPPH radical (Figure 4). The results on the optical absorption are almost similar, except that curcumin and mainly quercetin, which were used as an antioxidant standard, had a more pronounced effect than NDS27 and the curcumin salt.

The enzymes are the key players of the ROS/free radicals in the living systems. Understanding their interaction with the drug candidate could be a good approach. The peroxidase enzyme is known to generate the radical π–cation and radical centered on the porphyrin and can amplify the oxidant redox state of the milieu being studied [61]. In humans, myeloperoxidase has been widely studied, and its interaction with curcumin has also been explored [23]. The horseradish peroxidase (HRP) found in plants is also widely studied to predict and select new candidates for antioxidant activity, as well as for biochemical assays [62,63]. Herein, we have used this enzyme to better understand the mechanism of action of the curcumin complex (NDS27).

The chemiluminescence test is based on the reaction of oxidizing radicals derived from the molecules with probes to produce the excited-state species that emit light. Any compound that can react with radical initiators might inhibit or enhance the production of light. We have used this technique to study the antioxidant activity of the studied compounds to better understand their mechanism of action towards the enzymatic system, which is involved in the redox phenomenon and inflammation. The enzyme-catalyzed oxidation model using horseradish peroxidase (HRP) was employed based on the literature data [63,64]. Herein, we have used HRP-H_2_O_2_ as a source of radical cation and L012 as a chemiluminescent probe. HRP reacts with hydrogen peroxide to produce reactive species such as cation radical intermediate [P-Fe ^IV = 0^]^•+^, also named compound I (see Figure 2). The latter reacts with L-012 to produce an excited intermediate and the resulting light emission. Thus, in the presence of an antioxidant molecule, the latter is oxidized by compound I, thereby inhibiting the luminescence.

Using optical absorption, we have shown that NDS27 slightly enhances the formation of reactive intermediates of HRP. These results are confirmed, using the CL technique, by the nice dose-dependent inhibitory action of NDS27, curcumin, and the salt under lysine form. To better understand the mechanism of action, an EPR study was conducted with or without the addition of a spin trap agent. Like the DPPH assay, a single-line EPR signal was observed when the HRP/H_2_O_2_ system is used in the absence of spin trap DMPO, indicating that curcumin, which is encapsulated in the cyclodextrin, still has the capability to exchange with the porphyrin radical, either through a hydrogen atom transfer (HAT) or electron transfer (ET) mechanism.

We also observed that this porphyrin radical state does not last and that the EPR signal disappears. Based on these results, we have used a docking study to obtain more insight and understand the experimental results obtained from the HRP peroxidase inhibitory assay. As previously mentioned, the docking results have shown that curcumin can be inserted into the hydrophobic binding pocket of the HRP and interact with various residues in the domain. Indeed, the HRP binding domain is described as a cavity formed by protoporphyrin heme (C_34_H_32_FeN_4_O_4_), acetate ion, and Arg31, Ala34, Ser35, Arg38, Phe41, Ser73, Pro139, Ala140, Pro141, Phe152, Leu166, Ser167, Gly169, His170, Phe172, Gly173, Lys174, Asn175, Gln176, Phe179, Phe221, and Ser246 amino acid residues. Therefore, the interactions of curcumin with amino acids Arg38, His42, Ser73, Ser151, Ser167, His170, and Asp247, with an interatomic “separating” distance of 1.4, 1.6, 2.9, 2.4, 2.4, 1.6, and 1.1 Å, respectively, suggest that the contacts that could block the peroxidase activity can explain our results. The separating distance of each of these putative interactions is presented in Appendix A. These interactions act by modifying the binding domain native activity mechanism. The best-docked pose binding energy was also obtained to understand the ligand–receptor affinity. The free binding energy found from the docking is fair, with a value of –1.23 kcal/mol. The van der Waals + hydrogen bonding + de-solvation energy for the curcumin–HRP complex was −1.31 kcal/mol, i.e., seven times higher than the positive electrostatic energy value of +0.08 kcal/mol. This result suggests that hydrogen bonding and van der Waals interaction are the major forces stabilizing the docked complex. The value obtained shows the low stabilization of the curcumin–HRP complex since the molecular docking simulations could be considered practical when the binding energy was ~ or <−1.2 kcal/mol [65,66].

## 5. Conclusions

Numerous in vitro and in vivo studies have been devoted to studying the pharmacological effects of curcumin. Amongst them, antioxidant and anti-inflammatory properties have been widely studied. Curcumin is a multitarget drug with efficacy for multiple chronic diseases; however, poor solubility and low bioavailability have led to the development of several new formulations with specificity for the target compartment. The use of adjuvants (liposomes, nanoparticles, phospholipids complexes, various oils, etc.) improves those properties. In this study, we present another approach to increase the curcumin solubility and bioavailability and explore the free radical’s inhibition of the curcumin complex, named NDS27. Various strategies are used in this work to assess the curcumin radical scavenging effect through EPR and chemiluminescence. Our findings indicate that NDS27 enhances the curcumin solubility and remains as efficient as curcumin on various free radicals, such as DPPH^•^/ABTS^•+^, hydroxyl, superoxide anion, and lipid peroxyl species. In addition, NDS27 can quench singlet oxygen and inhibit horseradish peroxidase-catalyzed oxidation. The interactions of the curcumin released by NDS27 with the peroxidase active site were explored through a docking simulation. Altogether, our findings, using a combination of EPR spectroscopy with complementary techniques, provide insights into the mechanism of action of the new water-soluble curcumin form (NDS27) which, besides the improvement of its solubility and bioavailability, maintains an activity that is similar to curcumin. Because of the similar chemical structure, the new curcumin formulation, NDS27, with cyclodextrin as an adjuvant, exhibits scavenging activity against ROS and free radicals through a radical transfer process, as the electron can be delocalized along the entire structure. These interesting properties and mechanisms of action might be useful for better pathology targeting when using this new formulation for the treatment of inflammatory diseases.

## Data Availability

Data is contained within the article.

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
