# Peer review of "Free Radical Inhibition Using a Water-Soluble Curcumin Complex, NDS27: Mechanism Study Using EPR, Chemiluminescence, and Docking"

_antioxidants, 2024, doi:10.3390/antiox13010080_

Round 1
Reviewer 1 Report
Comments and Suggestions for Authors
The submitted manuscript describes the results of the combined experimental (EPR, biological measurements) and theoretical (very basic molecular docking) approaches to study the free radical inhibition by the complex of lysine, curcumin and HP-BCD, obtained previously by the same authors. While the amount of newly presented data is large, the work also requires revision.
The chemical structure of NDS27 should be presented in the introduction.
Figure 1, the Authors should have used statistical methods (i.e. ANOVA) to compare the results.
Line 88, here, it should be stated that apart from the EPR method, used in this study, also NMR can be used as an alternative method to verify the quality of dietary supplements containing curcumin
Line 328, molecular docking is a very basic approach. To get more insight the authors should have performed the molecular dynamics simulations and MM/GBSA calculations.
Molecular docking: the whole study is devoted to NDS27, which is a combination of two guests, curcumin and lysine. Therefore, I don’t understand, why for the docking to HRP solely the curcumin molecule has been used? It indicates that the effect of curcumin and NDS27 effect should be the same.
Lines 781-784, those are neither results nor conclusions
Molecular docking: why the Authors haven’t provided any values, i.e. the energy values, but solely the figures?
Figures S1 and S4, why there are only “up” error bars?
Figure S6E, it should be “time”
Figure S6, there is inconsistency in the error bars, in some figures there are both sides and also vertical plus horizontal, in the other figures there are none. Please correct.
Author Response
"Please see the attachment"

Reviewer 2 Report
Comments and Suggestions for Authors
1. This paper used a water-soluble curcumin complex, NDS27, to address the solubility and bioavailability of curcumin. However, the current description of the experimental setup lacks a detailed methodology, especially in the analysis of free radical scavenging and enzyme-catalyzed oxidation. It is important to include the exact conditions under which the experiments were performed, such as the concentration of reagents, reaction time, and temperature. In addition, it is necessary to present detailed data, including numerical and statistical analysis.
2. You also stated that you developed NDS27 to overcome the solubility and bioavailability problems of curcumin. However, a comparative study of other existing curcumin complexes is needed. Present the results of the comparison and explain its effectiveness and advantages.
3. While this study discusses in vitro studies, it would be useful to include in vivo studies to evaluate the bioefficacy and safety profile of NDS27. These could include evaluating the compound's bioavailability, inflammation, efficacy, and potential toxicity.
4. In the presented study, the authors briefly described a molecular docking study to investigate the interaction of NDS27 with the active site of the peroxidase enzyme. It would be helpful to provide more information on the molecular interactions, binding affinities, and potential effects on enzyme activity. Also discuss the potential mechanisms by which NDS27 exerts its antioxidant and anti-inflammatory effects.
5. The authors are addressing the solubility of curcumin through the development of NDS27. The authors present results on the solubility properties of NDS27 compared to natural curcumin, including solubility in various solvents and at various pH levels. Include a detailed discussion.
Comments on the Quality of English Language
Kindly review the paper meticulously for correct use of English. Although the overall English quality aligns with the standards of the journal, there are specific areas where clarity could be furtherly enhanced by rewording certain phrases and improving the syntax.
Author Response
"Please see the attachment"

Round 2
Reviewer 1 Report
Comments and Suggestions for Authors
The Authors have sufficiently improved their work. This version is accceptable.
Reviewer 2 Report
Comments and Suggestions for Authors
The authors have improved their manuscript on a number of important issues. Now this paper could be accepted in this journal.
Comments on the Quality of English LanguageNo comments.